# Unveiling Species Diversity Within Early-Diverging Fungi from China VII: Seven New Species of *Cunninghamella* (*Mucoromycota*)

**DOI:** 10.3390/jof11060417

**Published:** 2025-05-29

**Authors:** Zi-Ying Ding, Meng-Fei Tao, Xin-Yu Ji, Yang Jiang, Yi-Xin Wang, Wen-Xiu Liu, Shi Wang, Xiao-Yong Liu

**Affiliations:** 1College of Life Sciences, Shandong Normal University, Jinan 250358, China; 15270343451@163.com (Z.-Y.D.); taomengfei0808@163.com (M.-F.T.); ji15965902393@163.com (X.-Y.J.); jiangyang202309@126.com (Y.J.); wyx13953348060@163.com (Y.-X.W.); 18054356852@163.com (W.-X.L.); wangssdau@126.com (S.W.); 2Institute of Microbiology, Chinese Academy of Sciences, Beijing 100101, China

**Keywords:** *Mucorales*, basal fungi, fungal diversity, taxonomy, molecular phylogeny

## Abstract

The fungal genus *Cunninghamella*, ubiquitously distributed as saprotrophic organisms with occasional endophytic and phytopathogenic manifestations, holds significant biomedical and biochemical importance. During systematic investigations of fungal biodiversity in China, seven novel taxa (*C. amphispora* sp. nov., *C. cinerea* sp. nov., *C. flava* sp. nov., *C. hainanensis* sp. nov., *C. rhizoidea* sp. nov., *C. simplex* sp. nov., and *C. yunnanensis* sp. nov.) were delineated through integrated taxonomic approaches incorporating morphological characterization, multilocus phylogenetic analyses (ITS-LSU-*TEF1α*), and physiological assessments. Phylogenetic reconstructions positioned these novel species within a well-supported clade along with *C. bainieri* and *C. verticillata*. New species and their diagnostic features are *C. amphispora*, exhibiting dimorphic sporangiola production; *C. cinerea*, distinguished by gray pigmentation in the colony; *C. flava*, characterized by a yellow colony; *C. hainanensis* and *C. yunnanensis,* bearing eponymous designations reflecting their geographic origins; and *C. simplex*, displaying simplified sporangiophore branching. Comprehensive taxonomic descriptions accompanied by photomicrographic illustrations are provided herein. This study constitutes the seventh installment in an ongoing series elucidating early-diverging fungal diversity in China, expanding the global *Cunninghamella* taxonomic inventory to 63 species and advancing our understanding of mucoralean phylogeny.

## 1. Introduction

The genus *Cunninghamella* belongs to *Mucoromycota*, *Mucoromycetes*, *Mucorales,* and *Cunninghamellaceae* [1]. It was established in 1930 by Matruchot and typified with *C. echinulata* (Thaxt.) Thaxt. ex Blakeslee [2,3]. Members of *Cunninghamella* are usually saprotrophs and sometimes endophytes, and they are ubiquitous globally in various environments, including soil, air, feces, and humus [4,5,6,7]. Meantime, some species of *Cunninghamella* are opportunistic pathogens causing infections in immunocompromised populations [6,8,9,10]. Additionally, *Cunninghamella* represents a pivotal contribution to the field of biomedicine and biochemistry, producing a rich array of metabolites such as fatty acids, terpenes, nickel-iron carriers, and sugars [11,12,13].

*Cunninghamella* species are characterized by the branching pattern of sporangiophores; the shape and size of terminal and lateral vesicles; and the shape, size, and surface of sporangiola [2,4,14,15,16,17,18]. In a monographic study of 2001, Zheng and Chen [2] delimited *Cunninghamella* species on the basis of morphological characteristics and maximum growth temperatures. Nevertheless, identification of certain *Cunninghamella* remains challenging due to considerable phenotypic variabilities [2]. Later, Yu et al. [10], Liu et al. [19], and Walther et al. [3] reached a consensus among morphological observations and molecular phylogenies of ITS rDNA (internal transcribed spacer of ribosomal DNA) and *TEF1α* (translation elongation factor 1 alpha), overcoming the limitations of traditional classification methods, thus providing a more accurate and comprehensive classification.

Currently, 56 *Cunninghamella* species are recorded in the Index Fungorum (http://www.indexfungorum.org/, accessed on 19 December 2024), and they are distributed in Antarctica, Australia, Belgium, Brazil, China, France, India, Japan, Korea, Papua New Guinea, Spain, Switzerland, Tanzania, Thailand, and the United States [2,7,10,14,16,17,20,21,22,23,24,25].

Seven novel species of *Cunninghamella* were discovered based on molecular, morphological, and physiological characteristics during the process of isolating fungi from soil samples collected from southern China. Phylogenetic trees, descriptions, and illustrations of these novel species are provided herein. This is the seventh report of a serial study on diversity of Chinese early-diverging fungi [26,27,28].

## 2. Materials and Methods

### 2.1. Sample Collection and Strain Isolation

In 2024, soil samples were collected in Yunnan province, China. Soil sample collection was initiated by removal of surface contaminants (including leaf litter, debris, and particulates) using a sterilized stainless steel shovel. Following surface preparation, intact soil spanning the 5–10 cm subsurface horizon was extracted. Approximately 500 g of homogenized soil material was then transferred into sterile sample bags. All collected samples were immediately labeled with waterproof markers indicating collection date, GPS coordinates (±3 m accuracy), and altitude, followed by temporary storage at 4 °C prior to laboratory processing [29,30]. Pure strains were isolated by a combination of the plate dilution coating method and the single spore separation method [31,32]. About 1 g of soil sample was added into a centrifugal tube containing 10 mL of sterile water. Then, a 10^−1^ soil suspension was obtained by shaking. One milliliter of the 10^−1^ soil suspension was transferred into a centrifuge tube containing 9 mL of sterile water to prepare a 10^−2^ soil suspension. In the same way, 10^−3^ and 10^−4^ soil suspensions were successfully obtained. A portion (200 µL) of 10^−3^ and 10^−4^ soil suspensions was poured on the surface of Rose Bengal. Chloramphenicol agar (RBC: peptone 5.00 g/L, glucose 10.00 g/L, MgSO_4_·7H_2_O 0.50 g/L, KH_2_PO_4_ 1.00 g/L Rose Bengal 0.05 g/L, chloramphenicol 0.10 g/L, agar 15.00 g/L) [33] containing 0.03% streptomycin sulfate, coated evenly with a glass coating rod, and then incubated in darkness at 25 °C for 3–5 days. When the colony grew, it was transferred onto potato dextrose agar (PDA: 200 g potato, 20 g dextrose, 20 g agar, 1000 mL distilled water, pH 7.0) for further cultivation. After sporangium formation, sporangiospores were dispersed by suspension with sterile water. The single spores were picked up by a sterile inoculation ring and transferred to a PDA plate. The plate was placed in an incubator at 25 °C for dark cultivation. All strains were stored at 4 °C in 10% glycerol. The holotype materials were deposited in the China General Microbiological Culture Collection Center, Beijing, China (CGMCC), and Shandong Normal University, Jinan, China (XG). The holotype specimens were deposited in the Herbarium Mycologicum Academiae Sinicae, Beijing, China (Fungarium, HMAS). Taxonomic information of these new strains was uploaded to the Fungal Names repository (https://nmdc.cn/fungalnames/, accessed on 25 May 2025).

### 2.2. Morphology and Maximum Growth Temperature

After dark culture on a PDA plate for 3–5 days, the phenotypic characteristics of fungi (obverse and reverse) were photographed using a high-definition color digital camera (DP80, Olympus, Tokyo, Japan). Micromorphological features of fungi were observed by a stereomicroscope (Olympus SZX10, OLYMPUS, Tokyo, Japan) and an optical microscope (BX53, Olympus, Tokyo, Japan). For the determination of the maximum growth temperature, all strains were initially cultured in the dark at 26 °C for 3 days, and then the temperature was gradually increased by 1 °C every day until reaching a maximum temperature of 38 °C on the 12th day [34,35,36].

### 2.3. DNA Extraction, PCR Amplification, and Sequencing

Genomic DNA was extracted from mycelia by BeaverBeads Plant DNA Kit (Cat. No.: 70409–20; BEAVER Biomedical Engineering Co., Ltd., Suzhou, China) [37,38]. Molecular markers ITS rDNA (internal transcribed spacer of ribosomal RNA gene), LSU rDNA (large subunit of ribosomal RNA gene), and *TEF1α* (translation elongation factor 1 alpha gene) were amplified by polymerase chain reaction (PCR), respectively, using the following primer pairs: ITS5 (5′-GGA AGT AAA AGT CGT AAC AAG G-3′)/ITS4 (5′-TCC TCC GCT TAT TGA TAT GC-3′) [39], LR0R (5′-GTA CCC GCT GAA CTT AAG C-3′)/LR5 (5′-TCC TGA GGG AAA CTT CG-3′) [40], and EF1-983F (5′- ATG ACA CCR ACR GCR ACR GTY TG-3′)/EF1-2218R (5′-AACTTGCAGGCAATGTGG-3′) [41]. The amplification procedure consisted of a predenaturation at 95 °C for 5 min, 35 cycles of denaturation at 95 °C for 30 s, annealing (ITS at 55 °C for 30 s, LSU at 52 °C for 30 s, and *TEF1α* at 56 °C for 1 min), and extension at 72 °C for 1 min, and an extra extension at 72 °C for 10 min. The amplification reaction was performed in a volume of 25 µL, comprising 12 µL 2 × Hieff Canace^®^ Plus PCR Master Mix (Yeasen Biotechnology, Shanghai, China, Cat No. 10154ES03), 10 µL ddH_2_O, 1 µL forward and reverse primers (10 µM) (TsingKe, Qingdao, China), and 1 µL fungal genomic DNA (about 1 µM). All PCR products were electrophoresed on 1% agarose gels to confirm specificity and stained with GelRed [42]. Gel recovery was performed using a Gel Extraction Kit (Cat# AE0101-C; Shandong Sparkiade Biotechnology Co., Ltd., Jinan, China). PCR products were sent to the Biosune Company Limited (Shanghai, China) for Sanger sequencing. Sequences were generated from both strands, aligned using MAFFT v.7.0, and assembled using MEGA v.7.0 [43,44,45]. ITS, LSU, and *TEF1α* sequences were submitted to GenBank for BLAST similarity searches (https://blast.ncbi.nlm.nih.gov/, accessed on 15 February 2025). All sequences newly acquired in this article were deposited at GenBank under the accession number in Appendix A.

### 2.4. Phylogenetic Analyses

Phylogenetic analyses were performed based on the combined sequences of ITS, LSU, and *TEF1α*. Sequences both generated herein and retrieved from NCBI were presented in Appendix A. Phylogenetic trees were reconstructed using maximum likelihood (ML) and Bayesian inference (BI) methods. ML analysis was performed using RAxML-HPC2 on XSEDE v.8.2.12 on the CIPRES Science Gateway website (https://www.phylo.org/, accessed on 15 February 2025) with 1000 bootstrap replicates under the GTRGAMMA model [46,47,48,49]. For BI analysis, a quick start configured with an automatic stop option was executed on a Linux system server [50,51,52]. Random initial trees were used to run six simultaneously running Markov chains for 5,000,000 generations, with samples being taken every 1000 generations. The first 25% of trees were discarded as burn-in, and the remaining trees were employed to measure the posterior probability (PP). The layout and adjustments of the phylogenetic tree were conducted on the iTOL website (https://itol.embl.de, accessed on 15 February 2025), and then the final beautification was performed through Adobe Illustrator CC 2019 (https://adobe.com/products/illustrator/, accessed on 15 February 2025).

## 3. Results

### 3.1. Phylogeny

The molecular dataset included 56 strains in *Cunninghamella*, with *Mucor janssenii* CBS 205.68 as an outgroup. The dataset consisted of 3989 characters, covering ITS rDNA (1–1692), LSU rDNA (1693–2912), and *TEF1α* (2913–3989). There were 2098 constant, 522 variable but parsimony-uninformative, and 1369 parsimony-informative characters. MrModelTest indicated that Dirichlet fundamental frequency along with the GTR + I + G evolution pattern was suitable for both partitions in Bayesian inference. The topology of the maximum likelihood (ML) evolutionary tree was highly congruent with that of the Bayesian inference (BI) evolutionary tree. Therefore, the ML evolutionary tree was selected as a representative for detailed illustration (Figure 1). Fourteen strains of *Cunninghamella* isolated in this study were grouped into seven independent clades. The new species *C. simplex* was closely related to *C. bainieri* with full support (MLBV = 100 and BIPP = 1.00) and then next to another new species, *C. cinerea* (MLBV = 100 and BIPP = 0.98). The new species *C. flava* had a close relationship with *C. verticillata*. (MLBV = 80 and BIPP = 0.95). The other four species each formed their own branches, that is *C. amphispora* (MLBV = 100, BIPP = 1), *C. hainanensis* (MLBV = 100 and BIPP = 1), *C. rhizoidea* (MLBV = 100 and BIPP = 0.99), and *C. yunnanensis* (MLBV = 100 and BIPP = 1).

### 3.2. Taxonomy

The descriptions and illustrations of seven novel *Cunninghamella* species are presented as follows:

#### 3.2.1. *Cunninghamella amphispora* Z.Y. Ding & X.Y. Liu, sp. nov., Figure 2

Fungal Names—FN 572727.

**Figure 2 jof-11-00417-f002:**
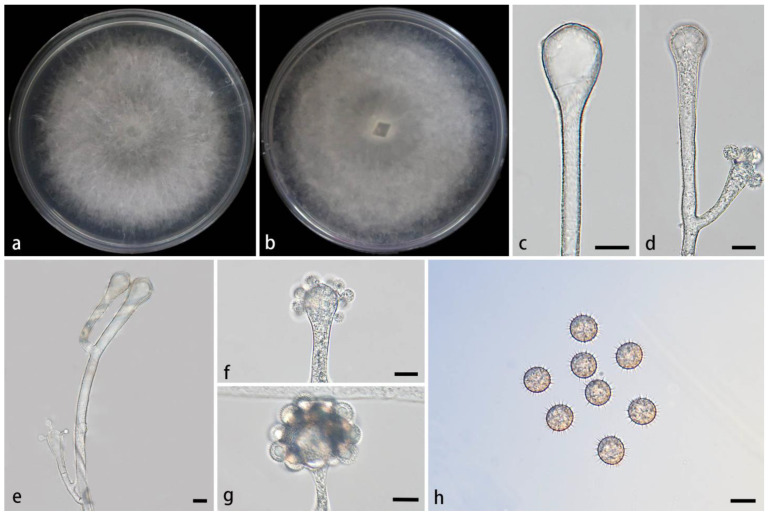
Morphologies of *Cunninghamella amphispora* ex-holotype CGMCC 3.28652. (**a**,**b**) Colonies on PDA, (**a**) obverse, (**b**) reverse; (**c**–**e**) sporangiophores with branching vesicles; (**f**,**g**) vesicles bearing sporangiola; and (**h**) echinulate sporangiola. Scale bars: (**c**–**h**) 10 μm.

Type—China, Yunnan Province, Puer City, Simao District, Yixiang Town, Dakai River section (22°60′26″ N, 101°09′54″ E, 1582.73 m), from soil, 5 July 2024, M.F. Tao and X.Y. Ji, holotype HMAS 353444, ex-holotype living culture CGMCC 3.28652 (=XG09634-9-1).

Etymology—The epithet *amphispora* (Lat.) refers to the species producing two types of sporangiola.

Description—Colonies on PDA at 26 °C for 4 days, reaching 85 mm in diameter, fast growing with a growth rate of 21.25 mm/d, initially white, gradually becoming light gray, floccose. Hyphae are branched, hyaline, smooth-walled, and aseptate when juvenile and septate when old. Rhizoids not found. Stolons are present. Sporangiophores arising from stolons or aerial hyphae, erect or few slightly bent, unbranched or simply branched, hyaline, single, never verticillate, always expanding upwards. They are 2.3–19.1 µm wide. Septa, if present, are single, usually just below the vesicles. Vesicles subglobose, globose, ovoid, or pillar-shaped, hyaline, smooth-walled, terminally 12.9–56.6 µm long and 11.6–46.7 µm wide, laterally 8–18.9 µm long and 5.1–14.8 µm wide. Pedicels over the entire surface of vesicles, 1.4–3.4 µm long. Sporangiola on pedicels, monosporous, globose, hyaline when young, dusky brown when old, thick-walled, 6.3–16.7 µm long, and 6.3–15.8 µm wide. Spines 0.5–2.6 µm long. Chlamydospores absent. Zygospores not found.

Maximum growth temperature—29 °C.

Additional strains examined—China, Yunnan Province, Puer City, Simao District, Yixiang Town, Dakai River section (22°60′26″ N, 101°09′54″ E, altitude 1582.73 m), from a soil sample, 5 July 2024, M.F. Tao and X.Y. Ji, living culture XG09634-9-2.

GenBank accession numbers—CGMCC 3.28652 (ITS, PV089203; LSU, PV123104; and *TEF1α*, PV200769), XG09634-9-2 (ITS, PV089204; LSU, PV123105; and *TEF1α*, PV200770).

Notes—In the phylogenetic tree of ITS-LSU-*TEF1α*, two strains of the *Cunninghamella amphispora* sp. nov. formed a fully supported independent clade (MLBV = 100, BIPP = 1.00; Figure 1), closely related to *C. cinerea* (BIPP = 0.97; Figure 1). These two species were distinguished by sporangiola, vesicles, pedicles, and rhizoids. The new species was distinguished from *C. cinerea* in producing one type of sporangiola (globose vs. globose and ovoid). The lateral vesicles were smaller than those of *C. cinerea* (18.9 µm × 14.8 µm vs. 20.3 µm × 17.3 µm). The pedicles of the new species were longer than those of *C. cinerea* (1.4–3.4 µm vs. 1.2–2.2 μm). Moreover, the new species produced abundant rhizoids, while *C. cinerea* was devoid of rhizoids. From a physiological perspective, the maximum growth temperature of the new species was lower than that of *C. cinerea* (29 °C vs. 33 °C).

#### 3.2.2. *Cunninghamella cinerea* Z.Y. Ding & X.Y. Liu, sp. nov., Figure 3

Fungal Names—FN 572721.

**Figure 3 jof-11-00417-f003:**
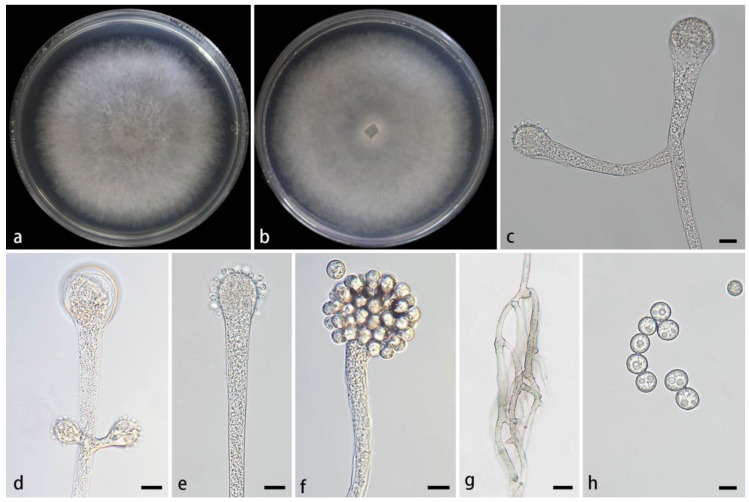
Morphologies of *Cunninghamella cinerea* ex-holotype CGMCC 3.28650. (**a**,**b**) Colonies on PDA: (**a**) obverse and (**b**) reverse; (**c**,**d**) sporangiophores with branching vesicles; (**e**,**f**) vesicles bearing sporangiola; (**g**) rhizoids; and (**h**) echinulate sporangiola. Scale bars: (**c**–**f,h**) 10 μm; (**g**) 20 μm.

Type—China, Yunnan Province, Puer City, Mojiang Hani Autonomous County, Tongguan Town (23°29′93″ N, 101°39′17″ E, 1471.21 m), from soil, 4 July 2024, M.F. Tao and X.Y. Ji, holotype HMAS 353442, ex-holotype living culture CGMCC 3.28650 (=XG09556-9-1).

Etymology—The epithet *cinerea* (Lat.) refers to gray colonies on PDA.

Description—Colonies on PDA at 26 °C for 4 days, reaching 85 mm in diameter, fast growing with a growth rate of 21.25 mm/d, initially white, gradually turning to smoky gray with age, and floccose. Hyphae are branched, hyaline, smooth-walled, and aseptate when juvenile and septate when mature. Rhizoids abundant, root-like, hyaline, and often branched. Stolons are present. Sporangiophores often arise from aerial hyphae, sometimes arising from stolons, erect or slightly bent, mainly unbranched or simply branched, hyaline, mainly single or recumbent, never verticillate, usually expanding upwards, 3.1–11.7 µm wide. Septa are rarely present, usually one or two below the vesicles in the main sporangiophores. Vesicles globose to elliptic, usually hyaline, smooth, terminally 14.3–32.3 µm long and 12.4–28.3 µm wide, and laterally 11.9–20.3 µm long and 9.9–17.3 µm wide. Pedicels covering the entire surface of vesicles, 1.2–2.2 µm long. Sporangiola on pedicles on vesicles, monosporous, globose to ovoid, hyaline when young, gradually dark brown when mature, thick-walled, 5.9–16.7 µm long, and 5.9–15.5 µm wide. Spines 0.8–1.7 µm long. Chlamydospores absent. Zygospores not found.

Maximum growth temperature—33 °C.

Additional strains examined—China, Yunnan Province, Puer City, Mojiang Hani Autonomous County, Tongguan Town (23°29′93″ N, 101°39′17″ E, altitude 1471.21 m), from soil, 4 July 2024, M.F. Tao and X.Y. Ji, living culture XG09556-9-2.

GenBank accession numbers—CGMCC 3.28650 (ITS, PV089197; LSU, PV123098; and *TEF1α*, PV172612), XG09556-9-2 (ITS, PV089198; LSU, PV123099; and *TEF1α*, PV172613).

Notes—Based on the ITS-LSU-*TEF1α* phylogenetic tree, two strains of the *Cunninghamella cinerea* sp. nov. formed a well-supported clade (MLBV = 100 and BIPP = 1.00; Figure 1), exhibiting a close relationship to *C. simplex* and *C. bainieri* (MLBV = 100 and BIPP = 0.98; Figure 1). Morphologically, the new species was distinguished from *C. simplex* by lateral vesicles (11.9–20.3 × 9.9–17.3 μm vs. 6.7–17 × 3.8–10.8 μm), shorter pedicles (1.2–2.2 μm vs. 1.1–2.9), and root-like rhizoids. The new species differed from *C. bainieri* by nonverticillate sporangiophores and shorter pedicles (1.2–2.2 μm vs. 2.5–3.5 (–6)). Besides, the new species had two shapes of sporangiola (globose and ovoid), but *C. bainieri* had four shapes (ovoid, ellipsoid, globose, and lacrymoid). Although both the new species and *C. bainieri* had rhizoids, the former was root-like and the latter was finger-like. Physiologically, the maximum growth temperature of the new species was 1 °C lower than that of *C. bainieri* (33 °C vs. 34 °C) [2].

#### 3.2.3. *Cunninghamella flava* Z.Y. Ding & X.Y. Liu, sp. nov., Figure 4

Fungal Names—FN 572722.

**Figure 4 jof-11-00417-f004:**
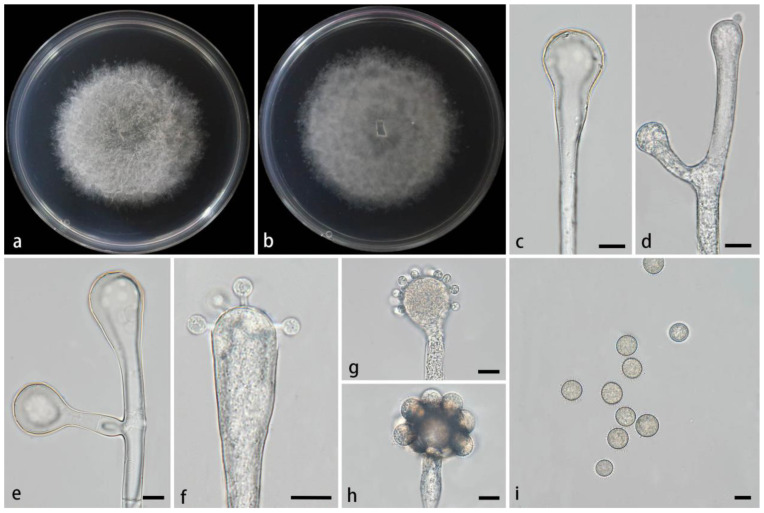
Morphologies of *Cunninghamella flava* ex-holotype CGMCC 3.28651. (**a**,**b**) Colonies on PDA, (**a**) obverse, (**b**) reverse; (**c**) sporangiophores; (**d**,**e**) sporangiophores with branching vesicles; (**f**–**h**) vesicles bearing sporangiola; and (**i**) echinulate sporangiola. Scale bars: (**c**–**i**) 10 μm.

Type—China, Yunnan Province, Puer City, Mojiang Hani Autonomous County, Tongguan Town (23°29′93″ N, 101°39′17″ E, 1471.21 m), from soil, 4 July 2024, M.F. Tao and X.Y. Ji, holotype HMAS 353443, ex-holotype living culture CGMCC 3.28651 (=XG9559-10-1).

Etymology—The epithet *flava* (Lat.) refers to the yellow colonies on PDA.

Description—Colonies on PDA at 26 °C for 6 days, reaching 72 mm in diameter, fast growing with a growth rate of 12 mm/d, initially white, gradually turning to dry yellow with age, floccose. Hyphae are branched, hyaline, smooth-walled, and aseptate when juvenile and septate when mature. Rhizoids are absent. Stolons are present. Sporangiophores arising from aerial hyphae or stolons, erect or slightly bent, unbranched or 1–7 branched, hyaline, single, recumbent, opposite, in pairs, 1–3 verticillate, sometimes growing irregularly, swelling or bulging, then tapering upward, 3.8–25.6 µm wide. Septa, if present, are usually one or two below the vesicles in the main sporangiophores. Vesicles globose, pear-shaped, or elliptic, usually hyaline and smooth, terminally 16.6–43.1 µm long and 15.4–44.5 µm wide, and laterally 9.3–28.1 µm long and 9.4–25.3 µm wide. Pedicels on the entire surface of the vesicle, 1.6–2.2 µm long. Sporangiola on pedicles, monosporous, globose to ovoid, hyaline when young, gradually tawny when mature, thick-walled, 8.9–18.6 µm long, 8.8–18.7 µm wide. Spines are 1.4–2.8 µm long. Chlamydospores are absent. Zygospores are not found.

Maximum growth temperature—33 °C.

Additional strains examined—China, Yunnan Province, Puer City, Mojiang Hani Autonomous County, Tongguan Town (23°29′93″ N, 101°39′17″ E, altitude 1471.21 m), from soil, 4 July 2024, M.F. Tao and X.Y. Ji, living culture XG09559-10-2.

GenBank accession numbers—CGMCC 3.28651 (ITS, PV089199; LSU, PV123100; and *TEF1α*, PV200765), XG09559-10-2 (ITS, PV089200; LSU, PV123101; and *TEF1α*, PV200766).

Notes—Based on the ITS-LSU-*TEF1α* phylogenetic tree, two strains of the *Cunninghamella flava* sp. nov. formed a fully supported lineage (MLBV = 100, BIPP = 1.00; Figure 1), closely related to *C. verticillata* (MLBV = 80 and BIPP = 0.95; Figure 1). However, they were different in terms of morphological features such as vesicles, pedicles, rhizoids, zygosporangia, and zygospores. The vesicles of the new species were characterized by various shapes, including globose, pear-shaped, and elliptic, while vesicles were quite regular in *C. verticillata*, mainly depressed-globose, subglobose to globose. The pedicles of the new species were shorter than those of *C. verticillata* (1.6–2.2 µm vs. 2.5–4 (–6.5) µm). Rhizoids were absent in the new species, whereas they were present in *C. verticillata*. Moreover, typical zygosporangia and zygospores were generated in *C. verticillata*, while absent in the new species. Physiologically, the maximum growth temperature of the new species was significantly lower than those of *C. verticillata* (33 °C vs. (37–) 39–42 °C) [2].

#### 3.2.4. *Cunninghamella hainanensis* Z.Y. Ding & X.Y. Liu, sp. nov., Figure 5

Fungal Names—FN 572723.

**Figure 5 jof-11-00417-f005:**
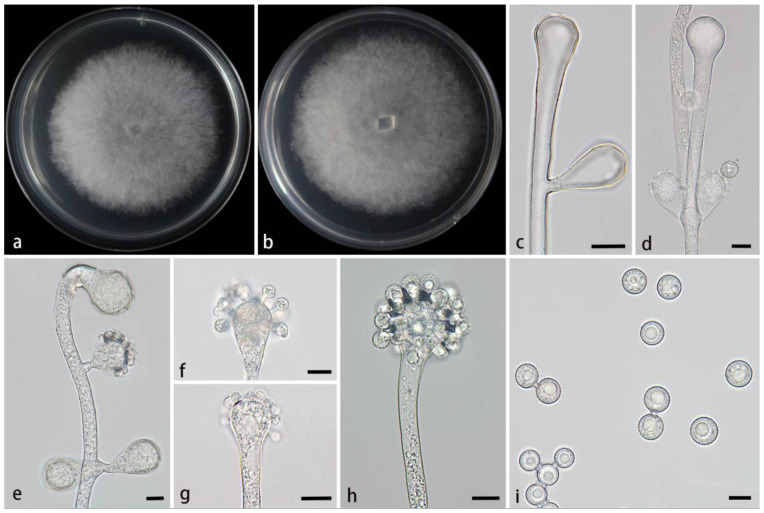
Morphologies of *Cunninghamella hainanensis* ex-holotype CGMCC 3.28649. (**a**,**b**) Colonies on PDA, (**a**) obverse, (**b**) reverse; (**c**–**e**) sporangiophores with branching vesicles; (**f**–**h**) vesicles bearing sporangiola; and (**i**) echinulate sporangiola. Scale bars: (**c**–**i**) 10 μm.

Type—China, Hainan Province, Danzhou City, near Hainan Tropical Botanical Garden (19°51′19″ N, 109°50′08″ E, 108 m), from a soil sample, 14 February 2024, M.F. Tao and X.Y. Ji, holotype HMAS 353441, ex-holotype living culture CGMCC 3.28649 (=XG06926-15-1).

Etymology—The epithet *hainanensis* (Lat.) refers to the location of the ex-holotype, Hainan Province, China.

Description—Colonies on PDA at 26 °C for 4 days, reaching 85 mm in diameter, fast growing with a growth rate of 21.25 mm/d, initially white, gradually becoming light gray, floccose. Hyphae are branched, hyaline, smooth-walled, and aseptate when juvenile and septate with age. Rhizoids are absent. Stolons are present. Sporangiophores arising from stolons or aerial hyphae, mostly erect, a few slightly bent, occasionally verticillate, hyaline, single, unbranched or 1–3 branched, opposite, in pairs, and 2.9–10.6 µm wide. Septa, if present, is usually one. Vesicles are subglobose, globose, or elliptic, smooth-walled, terminally 15.1–27.8 µm long and 11.7–26.8 µm wide, and laterally 16.8–28.4 µm long and 11.2–22.9 µm wide. Pedicels on the entire surface of vesicles, 1.2–5.8 µm long. Sporangiola are borne on pedicles, monosporous, mostly globose, hyaline when young, dusky brown when old, thick-walled, 7.3–14 µm long, and 7.8–13.4 µm wide. Spines are 0.7–1.9 µm long. Chlamydospores are absent. Zygospores are not found.

Maximum growth temperature—29 °C.

Additional strains examined—China, Hainan Province, Danzhou City, near Hainan Tropical Botanical Garden (19°51′19″ N, 109°50′08″ E, altitude 108 m), from a soil sample, 14 February 2024, M.F. Tao and X.Y. Ji, living culture XG06926-15-2.

GenBank accession numbers—CGMCC 3.28649 (ITS, PV089195; LSU, PV123096; and *TEF1α*, PV172610), XG06926-15-2 (ITS, PV089196; LSU, PV123097; and *TEF1α*, PV172611).

Notes—Based on the ITS-LSU-*TEF1α* sequence, two strains of the *Cunninghamella hainanensis* sp. nov. formed a distinct clade with full support (MLBV = 100 and BIPP = 1.00; Figure 1), showing a close relatedness to *C. rhizoidea* (BIPP = 0.99; Figure 1). Morphological comparisons revealed notable differences between these two new species in spine length and rhizoids. The spines of *C. hainanensis* were shorter than those of *C. rhizoidea* (0.7–1.9 µm × 1.3–2.0 µm). Furthermore, *C. hainanensis* possessed numerous rhizoids, while the feature was absent in *C. rhizoidea.* Physiologically, *C. hainanensis* demonstrated a significantly lower maximum growth temperature than that of *C. rhizoidea* (29 °C vs. 33 °C).

#### 3.2.5. *Cunninghamella rhizoidea* Z.Y. Ding & X.Y. Liu, sp. nov., Figure 6

Fungal Names—FN 572724.

**Figure 6 jof-11-00417-f006:**
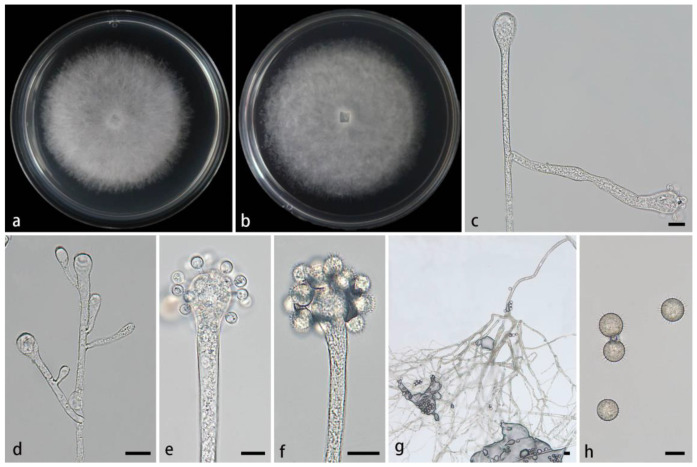
Morphologies of *Cunninghamella rhizoidea* ex-holotype CGMCC 3.28654. (**a**,**b**) Colonies on PDA, (**a**) obverse, (**b**) reverse; (**c**,**d**) sporangiophores with branching vesicles; (**e**,**f**) vesicles bearing sporangiola; (**g**) rhizoids; and (**h**) echinulate sporangiola. Scale bars: (**c**–**f**,**h**) 10 μm; (**g**) 20 μm.

Type—China, Yunnan Province, Lincang City, Gengma Dai Autonomous County, He Pai Township (23°48′32″ N, 99°41′66″ E, 988.59 m), from soil, 7 July 2024, M.F. Tao and X.Y. Ji, holotype HMAS 353446, ex-holotype living culture CGMCC 3.28654 (=XG09702-9-1).

Etymology—The epithet *rhizoidea* (Lat.) refers to the species producing abundant rhizoids.

Description—Colonies on PDA at 26 °C for 4 days, reaching 85 mm in diameter, fast growing with a growth rate of 21.25 mm/d, initially white, gradually becoming gray with age and floccose. Hyphae are branched, hyaline, smooth-walled, and aseptate when adolescent and septate when mature. Rhizoids are obviously present, root-like, complexly branching, and abundant. Stolons are present. Sporangiophores are formed from aerial hyphae or stolons, erect or slightly bent, mainly simple branches or occasionally multiple branches, hyaline, single, recumbent, opposite, occasionally verticillate, sometimes having a swollen area below the sporangiophores, and 2.8–11.4 µm wide. Septa, if observed, are usually only one in sporangiophores. Vesicles are globose, club-shaped, hyaline, smooth, terminally 9.3–35.3 µm long and 9.2–32.0 µm wide, and laterally 2.7–26.6 µm long and 2.4–27.1 µm wide. Pedicels are formed on the surface of the vesicle, 1.5–5.4 µm long. Sporangiola are formed on pedicles on vesicles, monosporous, globose to ovoid, hyaline when young, gradually tea brown when mature, thick-walled, 8.2–13.4 µm long, 7.9–12.9 µm wide, with short spines. Spines are 1.3–2.0 µm long. Chlamydospores are absent. Zygospores are not found.

Maximum growth temperature—33 °C.

Additional strains examined—China, Yunnan Province, Puer City, Mojiang Hani Autonomous County, Tongguan Town (23°29′93″ N, 101°39′17″ E, altitude 1471.21 m), from soil, 4 July 2024, M.F. Tao and X.Y. Ji, living culture XG09702-9-2.

GenBank accession numbers—CGMCC 3.28654 (ITS, PV089205; LSU, PV123106; and *TEF1α*, PV222157), XG09702-9-2 (ITS, PV089206; LSU, PV123107; and *TEF1α*, PV222158).

Notes—Based on the analysis of the ITS-LSU-*TEF1α* sequence, two strains of the *Cunninghamella rhizoidea* sp. nov. formed a fully supported independent lineage (MLBV = 100 and BIPP = 0.99; Figure 1), exhibiting close genetic relatedness to *C. hainanensis* (BIPP = 0.99; Figure 1). Morphological comparisons further highlighted significant differences between the two species in terms of sporangiola, spine length, and rhizoids (see the note to *C. hainanensis* above).

#### 3.2.6. *Cunninghamella simplex* Z.Y. Ding & X.Y. Liu, sp. nov., Figure 7

Fungal Names—FN 572725.

**Figure 7 jof-11-00417-f007:**
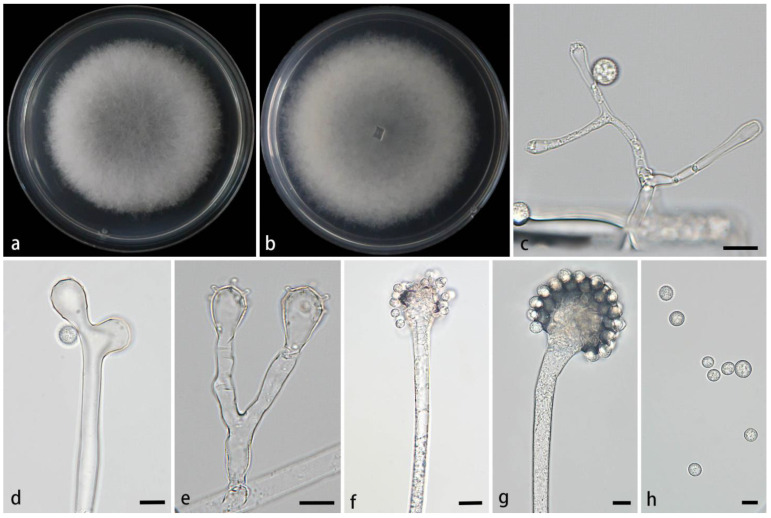
Morphologies of *Cunninghamella simplex* ex-holotype CGMCC 3.28653. (**a**,**b**) Colonies on PDA, (**a**) obverse, (**b**) reverse; (**c**–**e**) sporangiophores with branching vesicles; (**f**,**g**) vesicles bearing sporangiola; and (**h**) echinulate sporangiola. Scale bars: (**c**–**h**) 10 μm.

Type—China, Yunnan Province, Puer City, Simao District, Yixiang Town, Yutang Section, Longlongba Jinyu Tea Estate (22°68′48″ N, 101°07′32″ E, 1549.97 m), from soil, 5 July 2024, M.F. Tao and X.Y. Ji, holotype HMAS 353445, ex-holotype living culture CGMCC 3.28653 (=XG09611-12-1).

Etymology—The epithet *simplex* (Lat.) refers to the simple branching pattern of sporangiophores.

Description—Colonies on PDA at 26 °C for 4 days, reaching 85 mm in diameter, fast growing with a growth rate of 21.25 mm/d, initially white, gradually turning to dusky gray, and floccose. Hyphae are branched, hyaline, smooth-walled, and aseptate when young and septate with age. Rhizoids are absent. Stolons are present. Sporangiophores arising from stolons or aerial hyphae, erect or slightly bent, hyaline, unbranched or simply branched, in pairs, never verticillate, and 3.1–12.2 µm wide. Septa are occasionally present below vesicles in the main sporangiophores, usually only one. Vesicles are globose, ovoid, or cylindrical, usually hyaline; sometimes light brownish; smooth; terminally 9.9–28.4 µm long and 9.0–25.8 µm wide; and laterally 6.7–17.0 µm long and 3.8–10.8 µm wide. Pedicels are over the surface of vesicles and 1.1–2.9 µm long. Sporangiola borne on pedicles, monosporous, globose to ovoid, hyaline when young, light brown with age, smooth, thick-walled, 5.9–10.2 µm long, and 5.6–9.7 µm wide. Spines are very short, sparse, and 0.3–0.8 µm. Chlamydospores are absent. Zygospores are not found.

Maximum growth temperature—33 °C.

Additional strains examined—China, Yinnna Province, Puer City, Simao District, Yixiang Town, Yutang Section, Longlongba Jinyu Tea Estate (22°68′48″ N, 101°07′32″ E, altitude 1549.97 m), from soil, 5 July 2024, M.F. Tao and X.Y. Ji, living culture XG09611-12-2.

GenBank accession numbers—CGMCC 3.28653 (ITS, PV089201; LSU, PV123102; and *TEF1α*, PV200767), XG09611-12-2 (ITS, PV089202; LSU, PV123103; and *TEF1α*, PV200768).

Notes—According to the ITS-LSU-*TEF1α* sequence, two strains of the *Cunninghamella simplex* sp. nov. constituted a distinct branch with full support (MLBV = 100 and BIPP = 1.00; Figure 1), closely related to *C. bainieri* (MLBV = 100 and BIPP = 1.00; Figure 1). These two species were obviously different in the morphology of sporangiophores, sporangiola, vesicles, pedicles, and rhizoids. The new species lacked pseudoverticillate and verticillate sporangiophores, while *C. bainieri* produced them. Additionally, the new species differed from *C. binariae* by the simply shaped sporangiola, globose to ovoid, in contrast to the ovoid to ellipsoid, globose, and lacrymoid of various sporangiola in *C. bainieri.* The difference between the new species and *C. bainieri* on vesicles was that the new species generates globose, ovoid, and cylindrical forms, whereas *C. bainieri* produced globose, subglobose, and ovoid shapes. The pedicles of the new species were shorter than those of *C. bainieri* (1.1–2.9 µm vs. 2.5–3.5 (–6) µm). The new species lacked rhizoids, while *C. bainieri* gave rise to finger-like rhizoids. Physiologically, the maximum growth temperature of the new species was one degree lower than that of *C. bainieri* (33 °C vs. 34 °C) [2].

#### 3.2.7. *Cunninghamella yunnanensis* Z.Y. Ding & X.Y. Liu, sp. nov., Figure 8

Fungal Names—FN 572726.

**Figure 8 jof-11-00417-f008:**
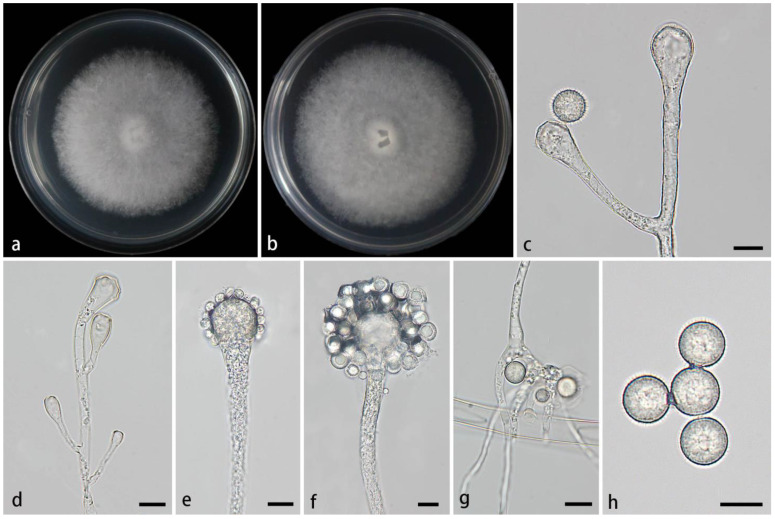
Morphologies of *Cunninghamella yunnanensis* ex-holotype CGMCC 3.28655. (**a**,**b**) Colonies on PDA, (**a**) obverse, (**b**) reverse; (**c**,**d**) sporangiophores with branching vesicles; (**e**,**f**) vesicles bearing sporangiola; (**g**) rhizoids; and (**h**) echinulate sporangiola. Scale bars: (**c**–**h**) 10 μm.

Type—China, Yunnan Province, Baoshan City, Longyang District, Lujiang Dam (24°92′66″ N, 98°88′15″ E, 703.72 m), from soil, 10 July 2024, M.F. Tao and X.Y. Ji, holotype HMAS 353447, ex-holotype living culture CGMCC 3.28655 (=XG10042-9-1).

Etymology—The epithet *yunnanensis* (Lat.) refers to the location, Yunnan Province, China, where the ex-holotype was collected.

Description—Colonies on PDA at 26 °C for 4 days, reaching 85 mm in diameter, fast growing with a growth rate of 21.25 mm/d, initially white, gradually becoming light gray with age, and floccose. Hyphae are branched, hyaline, smooth-walled, exuberant, and aseptate when juvenile and septate when old. Rhizoids are present, root-like, hyaline, and rarely branched. Stolons are present. Sporangiophores produce from stolons or aerial hyphae, erect or slightly bent, mainly unbranched or occasionally 1–7 branched, hyaline, straight or recumbent, few verticillate, usually expanding upwards, and 1.8–14.6 µm wide. Septa are sometimes one or two below the vesicles in the main sporangiophores. Vesicles are subglobose, globose, ovoid, elliptic, hyaline, smooth, terminally 10.5–33.8 µm long and 9.6–33.5 µm wide, and laterally 5.5–13.9 µm long and 4–12.4 µm wide. Pedicels are present throughout the vesicles and 1.1–4.1 µm long. Sporangiola are borne on pedicles, monosporous, mainly globose, sometimes ovoid, hyaline when young, grown when mature, thick-walled, 5.2–12.4 µm long, and 4.1–12.0 µm wide, with short spines. Spines are 0.7–2.0 µm long. Chlamydospores are absent. Zygospores are not found.

Maximum growth temperature—33 °C.

Additional strains examined—China, Yunnan Province, Baoshan City, Longyang District, Lujiang Dam (24°92′66″ N, 98°88′15″ E, altitude 703.72 m), from soil, 10 July 2024, M.F. Tao and X.Y. Ji, living culture XG10042-9-2.

GenBank accession numbers—CGMCC 3.28655 (ITS, PV089207; LSU, PV123108; and *TEF1α*, PV222159), XG10042-9-2 (ITS, PV089208; LSU, PV123109; and *TEF1α*, PV222160).

Notes—Based on the ITS-LSU-*TEF1α* phylogenetic tree, two strains of the *Cunninghamella yunnanensis* sp. nov. formed a fully supported independent lineage (MLBV = 100 and BIPP = 1.00; Figure 1), closely related to *C. hainanensis* (MLBV = 90 and BIPP = 0.98; Figure 1). They differed significantly in sporangiola, vesicles, pedicle length, and rhizoids. The sporangiola of the new species were smaller than those of *C. hainanensis* (5.2–12.4 × 4.1–12.0 µm vs. 7.3–14 × 7.8–13.4 µm). The maximum terminal vesicles of the new species were larger (33.8 × 33.5 µm vs. 27 × 26.5 µm), and the lateral vesicles were smaller than those of *C. hainanensis* (13.9 × 12.4 µm vs. 28.4 × 22.9 µm). The new species was shorter than *C. hainanensis* in pedicles (1.1–4.1 vs. 1.2–5.9 μm). Additionally, the new species possessed rhizoids, while *C. hainanensis* was absent. Physiologically, the maximum growth temperature of the new species was higher than that of *C. hainanensis* (33 °C vs. 29 °C).

## 4. Discussion

Currently, fungal taxonomists give paramount importance to molecular data when describing new taxa or evaluating interspecific relationships [53,54,55]. Specifically, sequences sourced from type materials are crucial for constructing a solid phylogenetic structure, serving as the foundation for a natural classification system [56]. Nonetheless, classical morphological characteristics and physiological characteristics continued to be widely acknowledged as a vital element in this process. The morphological and physiological characteristics and molecular data together provided a comprehensive picture of the species identity and evolutionary relationships [15,20]. In this study, seven novel species of *Cunninghamella* from southern China were identified based on morphological, physiological, and molecular data from type strains. And the morphological characteristics of these seven novel species and their relatives were systematically compared herein (Table 1).

The phylogenetic analysis of 14 strains using combined ITS-LSU-*TEF1α* sequences revealed 7 robust monophylies (Figure 1). *Cunninghamella hainanensis* and *C. rhizoidea* were sisters to each other. Morphologically, the former had shorter spines and more plentiful rhizoids. *C. cinerea* had a close relationship with *C. simplex* and *C. bainieri.* Compared with *C. simplex*, *C. cinerea* had larger lateral vesicles, shorter pedicels, rhizoids, and spineless sporangiola. *C. cinerea* differed from *C. bainieri* by nonverticillate sporangiophores, shorter pedicles, and different shapes of rhizoids and sporangiola [2]. In contrast to *C. verticillata*, which had almost all globular vesicles, the vesicles were more varied in *C. flava,* like globose, pear-shaped, and elliptic. Additionally, *C. flava* had shorter pedicels and no rhizoids. More importantly, *C. flava* lacked the classical sexual stage, which was present in *C. verticillata* [2]. In comparison to *C. cinerea*, *C. amphispora* had smaller lateral vesicles, longer pedicles, and spines. In comparison to *C. bainieri*, *C. simplex* had shorter pedicles, different shapes of vesicles, and sporangiola. Sporangiophores were not pseudoverticillate and verticillate in *C. simplex*, but they were present in *C. bainieri* [2].

In addition, temperature also served as a crucial factor in the categorization of fungi [14]. The maximum growth temperatures of the seven newly discovered species, *C. amphispora*, *C. hainanensis*, *C. cinerea*, *C. flava*, *C. rhizoidea*, *C. simplex,* and *C. yunnanensis,* were 29 °C, 29 °C, 33 °C, 33 °C, 33 °C, 33 °C, and 33 °C, respectively. Although the temperature tolerance of these new species was not very high, the maximum temperature of *C. verticillata*, a close relative of *C. flava*, was 42 °C [2]. *C. amphispora* and *C. hainanensis* were the same in maximum growth temperature but different in microscopic characteristics. In conclusion, both morphological and physiological traits were crucial for classifying fungi.

With the discovery of these seven new species, the genus *Cunninghamella* currently accommodates 63 species, among which 23 species were recorded in China. These species were mainly isolated from soil (19), faded flowers (2), endophytic in *Salicornia bigelovii* (1), and air (1) (http://www.indexfungorum.org/, accessed on 20 May 2025) [2,15,22,23,24,25]. From the perspective of the distribution of vegetation types, these species widely covered various ecological types in China, including the tropical monsoon forests and tropical rainforest areas in Hainan; the evergreen broad-leaved forests and tropical monsoon forests in the south subtropical zone of Guangdong; the evergreen broad-leaved forests in the subtropical zone of Jiangsu, Guizhou, Hubei, and Hunan; the deciduous broad-leaved forest in the warm temperate zone of Beijing; the temperate grassland and desert vegetation in Inner Mongolia; the temperate desert; and mountain vertical zone vegetation in Xinjiang. This cross-vegetation type distribution characteristic indicated that the species of the *Cunninghamella* genus had strong ecological adaptability and, at the same time, provided a new perspective for studying the correlation between fungal diversity and vegetation types.

## Figures and Tables

**Figure 1 jof-11-00417-f001:**
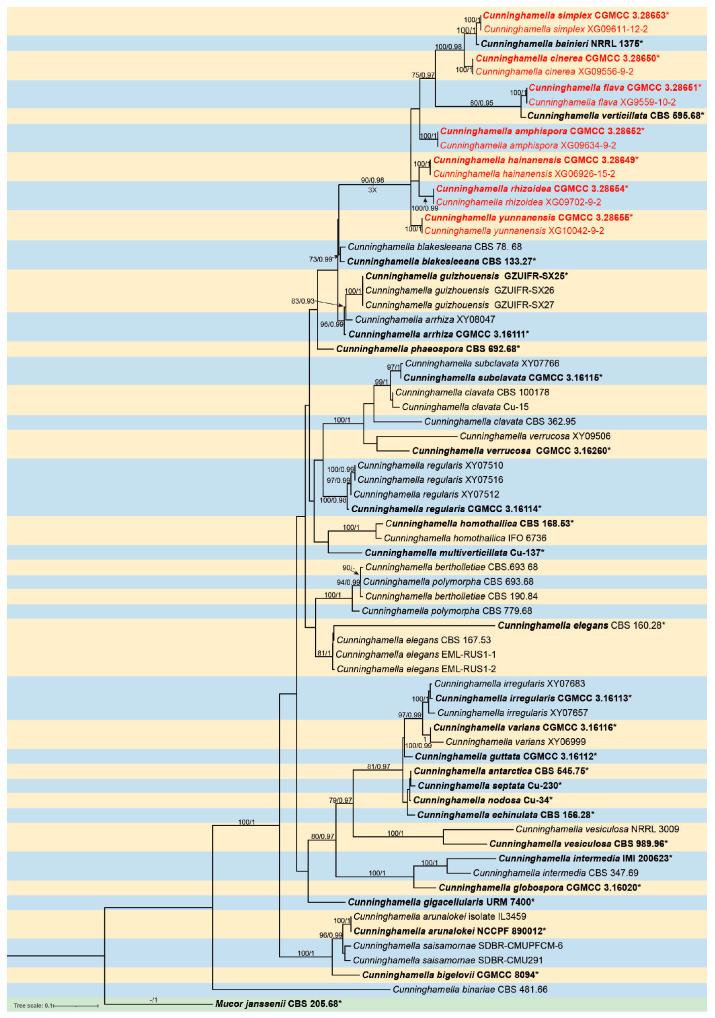
The maximum likelihood (ML) phylogenetic tree of *Cunninghamella* is based on sequences of ITS, LSU rDNA, and TEF1α, with *Mucor jansseni* as an outgroup. Nodes are annotated with ML bootstrap values (MLBV ≥ 70%) and BI posterior probabilities (BIPP ≥ 0.9), separated by slashes “/”. The bold black represents ex-type or ex-holotype strains with an asterisk “*”. Strains isolated in this study are displayed in red. The scale at the bottom left indicates 0.1 substitutions per site.

**Table 1 jof-11-00417-t001:** Morphological characteristics of *Cunninghamella* species.

Species	Colonies	Sporangiophores	Vesicles	Pedicels	Sporangiola	Reference
*C. hainanensis*	PDA: 26 °C 4 days, 85 mm, 21.25 mm/d, initially white, gradually becoming light gray, floccose	2.9–10.6 µm wide, mostly erect, a few slightly bent, occasionally verticillate, unbranched or 1–3 branched, opposite, in pairs	Globose, subglobose, elliptic, terminal vesicles 15.1–27.8 × 11.7–26.8 µm and lateral vesicles 16.8–28.4 × 11.2–22.9 µm	1.2–5.8 µm long	Mostly globose, 7.3–14 × 7.8–13.4 µm, with short spines, 0.7–1.9 µm long	This study
*C.* *cinerea*	PDA: 26 °C 4 days, 85 mm, 21.25 mm/d, initially white, gradually turning to smoky gray with age, floccose	3.1–11.7 µm wide, erect or slightly bent, mainly unbranched or simply branched, mainly single or recumbent, never verticillate	Globose to elliptic, terminal vesicles 14.3–32.3 × 12.4–28.3 µm and lateral vesicles 11.9–20.3 × 9.9–17.3 µm	1.2–2.2 µm long	Globose to ovoid, 5.9–16.7 × 5.9–15.5 µm, with short spines, 0.8–1.7 µm long	This study
*C. flava*	PDA: 26 °C 6 days, 72 mm, 12 mm/d, initially white, gradually turning to dry yellow with age, floccose	3.8–25.6 µm wide, erect or slightly bent, unbranched or 1–7 branched, recumbent, opposite, in pairs, 1–3 verticillate	Globose, pear-shaped, elliptic, terminal vesicles 16.6–43.1 × 15.4–44.5 µm and lateral vesicles 9.3–28.1 × 9.4–25.3 µm	1.6–2.2 µm long	Globose to ovoid, 8.9–18.6 × 8.8–18.7 µm wide, with short spines, 1.4–2.8 µm long	This study
*C.* *amphispora*	PDA: 26 °C 4 days, 85 mm, 21.25 mm/d, initially white, gradually becoming light gray, floccose	2.3–19.1 µm wide, erect or few slightly bent, unbranched or simply branched, hyaline, single, no verticillate	Subglobose, globose, ovoid, pillar-shaped, terminal vesicles 12.9–56.6 × 11.6–46.7 µm and lateral vesicles 8–18.9 × 5.1–14.8 µm	1.4–3.4 µm long	Globose, 6.3–16.7 × 6.3–15.8 µm, with short spines, 0.5–2.6 µm long	This study
*C. simplex*	PDA: 26 °C 4 days, 85 mm, 21.25 mm/d, initially white, gradually turning to dusky gray, floccose	3.1–12.2 µm wide, erect or slightly bent, hyaline, unbranched or simply branched, in pairs, never verticillate	Spherical, ovoid, cylindrical, terminal vesicles 9.9–28.4 × 9.0–25.8 µm and lateral vesicles 6.7–17.0 × 3.8–10.8 µm	1.1–2.9 µm long	Globose to ovoid, 5.9–10.2 × 5.6–9.7 µm, with very short spines, 0.3–0.8 µm long	This study
*C. rhizoidea*	PDA: 26 °C 4 days, 85 mm, 21.25 mm/d, initially white, gradually becoming gray with age, floccose	2.8–11.4 µm wide, erect or slightly bent, simple branches or occasionally multiple branches, recumbent, opposite, occasionally verticillate	Globose, club-shaped, terminal vesicles 9.3–35.3 × 9.2–32.0 µm and lateral vesicles 2.7–26.6 × 2.4–27.1 µm	1.5–5.4 µm long	Globose to ovoid, 8.2–13.4 × 7.9–12.9 µm, with short spines, 1.3–2.0 µm long	This study
*C. yunnanensis*	PDA: 26 °C 4 days, 85 mm, 21.25 mm/d, initially white, gradually becoming light gray with age, floccose	1.8–14.6 µm wide, erect or few slightly bent, mainly unbranched or occasionally 1–7 branched, straight or recumbent, few verticillate	Subglobose, globose, ovoid, elliptic terminal vesicles 10.5–33.8 × 9.6–33.5 µm and lateral vesicles 5.5–13.9 × 4–12.4 µm	1.1–4.1 µm long	Mainly globose, sometimes ovoid, 5.2–12.4 × 4.1–12.0 µm wide, with short spines, 0.7–2.0 µm long	This study
*C. bainieri*	SMA: 27 °C 4 days, 90 mm, at first white, soon becoming light gray, gray, to ‘Light Mouse Gray’, reverse cream, floccose	Erect, bent, or recumbent, main axes of sporangiophores (8–) 11–21 µm wide; primary branches (1–) 4–10 (–18) µm wide, monopodial, pseudoverticillate, or verticillate in 1–2 (–3) whorls of 3–8, typically in pairs	Globose, subglobose to ovoid, sometimes irregular, axial vesicles 18.5–32(–40) μm and lateral ones (8–)13.5–30 μm	2.5–3.5 (–6) μm long	Ovoid to ellipsoid and 7–14.5 (–20) × 6.5–11 (–14.5) µm globose and 5.5–12.5 µm, lacrymoid and 9–20 (–32.5) × 7–14.5 (–20) µm	[2]
*C. verticillata*	SMA: 28 °C 5–6 days, 90 mm, at first white, from the sixth day near ‘Avellaneous’ to near ‘Colonial Buff’, reverse yellowish cream, floccose	Erect, straight, or recumbent, main axes of sporangiophores 7.5–17.5 (–25) µm; branches (0–) 4–15 (–25), verticillate to pseudoverticillate, rarely singly or in pairs, mostly simple, very rarely re-branched	Axial ones slightly depressed—globose, subglobose to globose, 225–50 (–70) μm and lateral ones usually globose to subglobose, sometimes broadly ovoid, 8.5–27.5 (–32.5) μm	2.5–4 (–6.5) um long	Two kinds: globose, broadly ellipsoid ovoid and bluntly pointed at one end, 6–17.5 (–20) × 5.5–13.5 (–17.5) µm; dark giant sporangiola, globose, 11.5–17.5 (−25) µm	[2]

Notes: PDA—Potato Dextrose Agar, SMA—Synthetic *Mucor* Agar.

## Data Availability

The sequences from the present study were submitted to the NCBI database (https://www.ncbi.nlm.nih.gov/, accessed on 12 February 2025). The sequences were deposited in the GenBank database (Appendix A).

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
