# Peer review of "Unveiling Species Diversity Within Early-Diverging Fungi from China VII: Seven New Species of Cunninghamella (Mucoromycota)"

_jof, 2025, doi:10.3390/jof11060417_

Round 1

Reviewer 1 Report

The article presents a relevant contribution to the knowledge of the diversity of mucoral fungi in China; it uses a polyphasic approach in the descriptions of new species, and has good illustrations and taxonomic descriptions. However, before publication, it is necessary to correct typographical errors, review spelling and hyphenation, and make some adjustments. I recommend its publication after minor corrections.

Minor corrections:
• Line 3: Correct “Mucoromycota” to Mucoromycota (without incorrect hyphenation).
Abstract
• Line 18: "distinguished by distinctive gray pigmentation", pleonasm. Suggestion: “distinguished by gray pigmentation”.
Introduction
• Line 50: “United State”, correct to United States.
• Line 46: "and provided a more accurate and comprehensive way." replace with "thus providing a more accurate and comprehensive classification."

Materials and Methods
• Line 63: “sterial sample bags”, correct to sterile sample bags.
• Line 93–94: Temperature “increased by 1℃ every day” until growth ceased; it is important to indicate the maximum number of days/temperature reached, as the method may seem vague.
• Line 105–114: PCR parameters are reported, but it is missing whether they were verified by electrophoresis and whether the products were purified before sequencing, and whether this was done in both directions (reverse and forward).
Taxonomy
• Line 376: “Yinnna Province”, typo, correct to Yunnan Province.
Discussion
• Line 455: “forming new taxa or evaluating species interconnections” consider rewording to: “when describing new taxa or evaluating interspecific relationships”.
• Line 488–489: “lacked the classical sexual stage, but C. verticillata was present” Suggestion: “lacked the classical sexual stage, which is present in C. verticillata.”
• Line 490: “larger lateral vesicles, smaller lateral vesicles”, this seems to be contradictory information, please review.

• Insert the Fungal Names registration number (FN######) before publication.

Reviewer 2 Report

I recommended that you register with mycobank and receive a number.

I think there will be a comment on the name.

6 page 190 line, add more detailed or compare with other species.

15page 449 line, add more detailed or compare with other species.

Reviewer 3 Report

This is an excellent contribution and few other works provide information about this group

The photographs are of excellent quality; only the 6G should be considered

It is recommended to standardized the use of terms, for example, globose or sphaerical and not use them indiscrimitalely. Similarly, the term “spinose” is considered to be echinulate.

It is recommended to follow the Ulloa & Hanlin 2012 Dictionary

For the sizes of structures such as sporangia ang sporangiolas measurements should only be taken of the mature ones, because considering the smallest ones, the intervals become very wide and cause confusion

In the species, Cunninghamella amphispora; I hace my doubts about whether the species really has this condition of smooth, echinulate spores,  therefore, scanning electron microscopy should be considered.

The comparative table of species only includes those from the Clade; indicate the locality type and by vegetation type  those from Chine should be considered.
